# Microstructure and Wear Behavior of Heat-Treated Mg-1Zn-1Ca Alloy for Biomedical Applications

**DOI:** 10.3390/ma17010070

**Published:** 2023-12-22

**Authors:** Nuria Pulido-González, Sonia García-Rodríguez, Belén Torres, Joaquin Rams

**Affiliations:** Departamento de Matemática Aplicada, Ciencia e Ingeniería de Materiales y Tecnología Electrónica, Escuela Superior de Ciencias Experimentales y Tecnología, Universidad Rey Juan Carlos, C/Tulipán s/n, Móstoles, 28933 Madrid, Spain; nuria.pulido.gonzalez@urjc.es (N.P.-G.); sonia.garcia@urjc.es (S.G.-R.); joaquin.rams@urjc.es (J.R.)

**Keywords:** magnesium alloys, Mg-Zn-Ca alloys, sliding wear, heat treatment, precipitation, biodegradable implants

## Abstract

The microstructure and wear properties of a Mg-1wt.% Zn-1wt.% Ca (ZX11) alloy with different heat treatments have been investigated. The ZX11 alloy was tested in the as-cast state and after different heat treatment conditions: solution-treated (at 450 °C for 24 h), peak-aged (solution-treated + aged at 180 °C for 3 h), and over-aged (solution-treated + aged at 180 °C for 24 h). The microstructure of the as-cast sample showed a continuous intermetallic phase at the grain boundaries, while the heat-treated samples exhibited discrete precipitated particles within the grains. To evaluate the wear behavior, the samples were tested using a pin-on-disc configuration, where the wear rates and friction coefficients were measured at different loads and sliding speeds. An AZ31 magnesium alloy was used as the counterbody. The worn surfaces and the wear debris were studied to identify the main wear mechanisms corresponding to each test condition. The results indicated the presence of abrasion, oxidation, and adhesive wear mechanisms in all testing conditions. In the as-cast state, delamination and plastic deformation were the dominant wear mechanisms, while they were less relevant in the heat-treated conditions. The peak-aged samples exhibited the lowest wear rates, suggesting that modifying the distribution of intermetallic precipitates contributed to enhancing the wear resistance of the alloy.

## 1. Introduction

Magnesium alloys are interesting as materials with many potential applications. One of the main characteristics of Mg alloys is their low density, which is in the 1.75–1.85 g/cm^3^ range. They also have good specific mechanical properties because they combine moderate mechanical properties with their lightness, which makes Mg alloys key structural materials in portable electronics, sports equipment, household appliances, and the transportation sector. In all these fields, the use of light Mg alloys would help to reduce weight while maintaining the mechanical properties, which would reduce the strength required for many uses as well as the weight of vehicles, reducing the environmental impact of the transportation sector [1,2,3].

Magnesium is highly biocompatible and its mechanical properties, such as elastic modulus and yield strength, are closer to those of bones than the metallic alloys commonly used as load-bearing implants, i.e., 316 L stainless steel and Ti-6Al-4V [4]. The lower rigidity of Mg alloys reduces the risk of stress shielding, which is a phenomenon that could cause bone loss in the vicinity of the implant [5] and causes inadequate healing of the implanted zone. Also, Mg is a material that can be considered biodegradable as it dissolves in biological environments [6,7]. Concerning implants, Mg alloys avoid the need to remove the implant once the bone tissue has healed, reducing the number of surgeries needed and the risks associated with the surgical processes, as well as avoiding some problems that the implant would imply in the long term such as metallosis, corrosion-induced toxicity or allergic responses of the surrounding tissues, and problems to treat an infection in the implant, as it dissolves in the body.

One of the main considerations that are promoting the use of Mg alloys as biomaterials is the avoidance of alloying elements that do not appear naturally in the body. The biodegradable alloys dissolve in the body, which processes all its constituents. To prevent any harmful effects caused by the elements present in the alloy, it is preferable to use only elements that already appear in the body. One of the strategies for developing biocompatible Mg alloys consists of using Ca and Zn as the main alloying elements, which are essential for many body functions [8].

It has been reported that Ca improves both corrosion resistance in chloride solutions, like the human body fluid and blood plasma, and also the mechanical properties of Mg alloys [9]. Zn can improve castability and strength, although it is usually combined with other elements such as Ca, Mn, or Si [10]. In previous research, it was established that a Zn content of between 0.5 and 1.25–1.5 wt.% improved the corrosion behavior [11,12]. The combination of Ca and Zn provides alloys that can be age-hardened [13,14] showing improved corrosion resistance [15]. These behaviors are associated with the formation of homogeneously precipitated fine secondary phases in the α-Mg matrix. The most relevant ones are Ca_2_Mg_6_Zn_3_ and Mg_2_Ca, and their amount, distribution, and contribution to the hardening of the alloys depend on the alloy composition and the heat treatment applied [16].

On the other hand, the limited wear behavior of many Mg alloys restricts their use in demanding applications in the transportation and biomedical sector to situations in which contact between pieces can occur, as in gears, mechanisms, or multipart implants. Some studies have evaluated the wear behavior of Mg alloys with different compositions, but few of them establish the fundamental aspects of the wear mechanisms. Also, most of them are focused on the alloys used in the transportation sector [17,18] rather than on the biomedical one, so the wear conditions are different from one to another. The service conditions of biomaterials imply body temperature and contact with other tissues or other Mg alloys rather than contact between the Mg alloy studied and hard materials such as tool steel or carbides. This makes the use of conventional wear testing with hard materials such as steel not representative of the actual behavior of the Mg in the biomedical field. Therefore, there is a strong need of studying the behavior of the biodegradable Mg alloys when they are in contact with themselves or other Mg alloys.

Previous research has indicated that in the Mg-Ca-Zn alloys, as the Zn content increases from 3 wt.% to 4 wt.%, the mechanical properties and the wear resistance increase, due to the greater presence of intermetallic compound phases and the increase in the hardness of the alloy. The yield strength increases by 8.5% and the UTS by 8.1%. A minor difference is observed in the wear values [19]. However, it is important to note that, in this study, tests were conducted on two different material compositions with varying grain sizes under a single dry-wear condition, making it difficult to isolate the effect of microstructure from that of composition. Additionally, the wear mechanisms were not analyzed. Other authors investigated the influence of the heat treatments on the microstructure and corrosion behavior of Mg alloys with similar composition [20]. In this study, after aging, the maximum bending stress increases from 95 to 135 MPa in the ZX11 alloy. However, their impact on the wear behavior of these alloys has not been studied. Moreover, few studies analyze the effect of heat treatment on the wear behavior of biodegradable Mg-based alloys and only alloys such as the EV31A and Mg-Zn-Y-Zr have been studied [21,22]. Although these alloys are of interest, they have a high content of Gd (EV31A) and Zr (both), neither of them being a bioelement.

This study focuses on the use of the ZX11 alloy in which all the elements in its composition are bioelements and in which heat treatments can be used to modify the mechanical behavior including its hardness. The wear behavior of the ZX11 alloy under dry sliding conditions, employing the pin-on-disc test, has been used to identify how this alloy would evolve in the case of contact between two separate pieces of a Mg alloy, such as may occur in some applications and also in the case of partial dissolution of a Mg implant. Additionally, the impact of sliding speed and applied load on the wear rate and wear mechanisms of the heat-treated alloys has been studied. Therefore, the impact of the modification of microstructure, achieved through heat treatments, on the wear performance of the alloy has been determined.

## 2. Experimental Procedure

### 2.1. Materials

Mg-1Zn-1Ca magnesium alloy (hereafter ZX11) was supplied by Helmholtz Zentrum Geesthacht with the nominal composition shown in Table 1. This alloy has an equilibrated proportion of Zn and Ca, has adequate mechanical properties, and its microstructure can be strongly modified by applying heat treatments. Alloys were cast by permanent mold indirect chill casting. Mg ingots were melted under a protective atmosphere of argon with 0.2% SF_6_ and held at 720 °C. Then, Ca and Zn were added. The mixture was stirred for 5 min and cast into a mold. The details of the casting process can be found elsewhere [23,24].

### 2.2. Heat Treatments

As-cast ZX11 alloy was solution-treated at 450 °C for 24 h in an electric furnace (T4) and subsequently quenched into water to preserve the microstructure. Afterward, some samples were isothermally aged in an oil bath at 180 °C for different aging times (from 1 h to 24 h) (T6) [25]. The aging response with time was monitored using the Vickers microhardness test with a load of 0.1 kg and 20 s of dwell time. Ten individual indentations were made on each specimen to determine the average microhardness value. Table 2 shows the heat treatments followed in this study.

### 2.3. Wear Test

Wear tests were carried out at room temperature and under dry sliding conditions using a pin-on-disc configuration tribometer (Microtest MT/30), using the as-cast and heat-treated ZX11 alloy as pins, with a size of 15 × 2.5 × 2.5 mm. Pins were ground with different emery papers up to 1200 grit. Pins and counterbody surfaces were cleaned before the wear tests using isopropanol to avoid the presence of humidity and non-desirable films such as grease. At least three pins were tested for each wear condition.

The counterbody was an AZ31 magnesium alloy plate with the composition shown in Table 3. In most studies, steel was used as the counterbody, but this situation is not realistic for biomedical application. The most common contact situation would be to have a Mg alloy in contact with another piece of the same Mg alloy. This would involve using a ZX11 alloy pin and disc. However, it would not be possible to detect if there was transfer between the two samples in contact. For this reason, an AZ31 alloy was used to track material transfer. Both alloys show many similarities, but the AZ31 contains aluminum, unlike the tested alloy, which allows the material of the samples to be distinguished from that of the counterbody. ZX11 samples were used as pins in the tests, as there was less material available than AZ31.

The wear tests were carried out under loads of 2 N and 20 N, at sliding velocities of 0.05 m s^−1^ and 0.1 m s^−1^ for a sliding distance of 200 m. Before establishing the sliding distance, a larger test (5000 m) was performed which showed that a steady-state wear regime was achieved at 200 m of sliding distance; therefore, it acted as a control condition. The loads used are characteristic of the wear conditions in the possible uses of this alloy. The sliding speeds are higher than those of the application, but they need to be used in order to have an accelerated wear test. The wear testing machine continuously recorded the friction coefficient, wear depth, and sliding distance. Samples were weighed before and after the wear tests to determine the mass loss during the test. The wear rate was obtained by using the alloy density (Equation (1)). At least three samples were tested for each wear condition, and the statistical deviation was calculated to evaluate the stability of the wear condition evaluated.
(1)$$\frac{V}{L}=\frac{m/d}{L}$$
where *V* is the wear volume (m^3^), *L* is the sliding distance (m), *m* is the mass loss (kg), and *d* is the density of the alloy (kg/m^3^). The coefficient *V*/*L* is the wear rate.

To evaluate the wear rates of the material under different conditions, Archard’s law was applied (Equation (2)) [26]:(2)$$\frac{V}{L}=K\frac{W}{H}=kW$$
where *W* is the applied load (kgf), *H* is the hardness of the sample (kg/m^2^), *K* is Archard´s constant, and *k* is the specific wear rate. Both constants can be determined from the fitting of Equation (2).

This equation allows for evaluating the severity of wear, and strong changes in Archard’s constant or in the specific wear rate would indicate changes in the wear mechanisms, which must be verified by analyzing the worn surfaces of the pin and the counterbody.

### 2.4. Microstructural and Compositional Characterization

Microstructural and compositional investigations of the samples, the worn surfaces, and the debris were conducted by using a Scanning Electron Microscope (SEM, Hitachi S-3400 N, Tokyo, Japan) equipped with an Energy-Dispersive X-Ray Spectrometer (EDS, Bruker AXS Xflash Detector 5010, Billerica, MA, USA). For the identification of the precipitated phases, samples were etched in a Nital solution (2 vol.% nitric acid 98 vol.% ethanol).

## 3. Results and Discussion

### 3.1. Age-Hardening Response and Microstructure of the Alloys

Figure 1 illustrates the evolution of microhardness with aging time for the ZX11 alloy. The initial hardness value of the as-cast ZX11 alloy is approximately 62 HV. After solution treatment, this value decreases to around 56 HV. Subsequently, after undergoing an isothermal aging process at 180 °C for 3 h, the hardness reaches a value of about 71 HV. Beyond this point, the hardness decreases with further aging.

For the wear study, samples in four different conditions were tested (Table 2): as-cast state; solution-treated; solution-treated and subsequently aged for 3 h (referred to as peak-aged condition); and solution-treated and subsequently aged for 24 h (referred to as over-aged condition).

Figure 2 displays the microstructure of the various tested conditions of the ZX11 magnesium alloy. The constituent phases of this alloy in the as-cast condition were characterized in depth in a previous study [16]. In this condition (Figure 2a), the precipitated particles formed a quasi-continuous phase at the grain boundaries (GBs). Within this phase, two types of Zn-Ca precipitates with different contrast were observed: Ca_2_Mg_6_Zn_3_ (bright particles) and Mg_2_Ca (dark particles). These precipitates are commonly found in many Mg-Zn-Ca alloys, and for the ZX11 in the as-cast condition, the Mg_2_Ca appeared as discrete particles that were embraced by the Ca_2_Mg_6_Zn_3_ phase due to the solidification process. The Mg_2_Ca particles possess a higher melting point, and, during solidification, they were the first formed. Subsequently, the Ca_2_Mg_6_Zn_3_ phase precipitated around the binary phase, embracing these particles. Meanwhile, the Mg-matrix solidified and lost alloying elements from the precipitates, which finally formed at the GBs and resulted in the microstructure shown. Figure 3 shows an EDS mapping analysis of the Ca_2_Mg_6_Zn_3_ and Mg_2_Ca precipitates within the alloy.

The solution treatment (Figure 2b) caused the full dilution of the Ca_2_Mg_6_Zn_3_ phase, while only partially diluting the Mg_2_Ca particles, with solely the dark contrast phase being detectable. This difference is caused by the lower melting temperature of the Ca_2_Mg_6_Zn_3_ phase. The continuity of precipitated phases at GBs disappeared because they were provided by the ternary phase that dissolved. As a result, after the solution treatment, only the Mg_2_Ca precipitates appeared at GBs.

In the aged conditions, the general morphology exhibited similarities to the solution-treated state, but with larger precipitates. In the peak-aged condition (Figure 2c), the precipitates increased in size and number. In addition to the precipitated phases observed in the solution-treated condition, the microstructure of the alloy also revealed the presence of new particles within the grains resulting from the precipitation of the previously dissolved ternary phase. In the over-aged (Figure 2d) condition, the particles observed in the previous condition were retained. This indicates that most of the elements that were initially dissolved in the matrix had already precipitated around the particles formed during shorter heat treatment times. The presence of these particles in the over-aged condition suggests that the alloy had reached a state where the majority of the precipitates had formed and further aging did not significantly alter their composition or distribution. Hence, the slight decrease in hardness can be attributed solely to the enlargement of the grain size within the microstructure [20].

### 3.2. Wear Rate

The wear tests were conducted following the procedure and the configuration indicated in the experimental section, using a counterbody made of AZ31 magnesium alloy, while previous studies employ steel as the counterbody material. It is important to notice that in biomedical applications, it is not common to find Mg in direct contact with steel. Therefore, the steel–magnesium tribological system is not relevant to biomedical applications, apart from the fundamental knowledge that those studies provide. Continuous contact between materials with similar natures, such as two types of magnesium alloys, either identical or similar, is more common in real applications. In our study, the alloy selected as the counterbody is the AZ31 Mg alloy that is the most commonly used alloy in biomedical applications. Its composition, particularly due to the presence of Al, enables the identification of any material transfer from the counterbody to the tested samples. This selection ensures that the wear tests are closer to the biomedical scenarios.

Figure 4 shows the volumetric wear rates of the ZX11 alloy under different conditions after a sliding distance of 200 m. For most conditions, the application of heat treatments resulted in a reduction in the wear rate.

For a load of 2 N and a sliding speed of 0.05 m s^−1^ (Figure 4a), the wear rate was reduced by 32% for the solution-treated condition and by 60% for the peak-aged condition, compared to the as-cast state. However, the application of an over-aging treatment increased the wear rate, bringing it closer to the wear rate observed in the solution-treated condition, which was 32% lower than that of the as-cast samples.

For the 2 N load, the increase in the sliding speed to 0.1 m s^−1^ increased the wear rates (Figure 4b). The application of a solution treatment reduced the wear rate by 45% compared to the as-cast state. In the peak-aged condition, the wear rate was only slightly below (8%) that of the as-cast state. The application of an over-ageing treatment resulted in a 30% lower wear rate.

These findings indicate that for the 2 N load, heat treatments can effectively reduce the wear rate of the ZX11 alloy, with the peak-aged condition exhibiting the lowest wear rate for a sliding speed of 0.05 m s^−1^ and the solution-treated condition at a sliding speed of 0.1 m s^−1^.

For higher loads, i.e., 20 N, the evolution of the wear rates was different from that observed at lower loads (Figure 4a,b). At a sliding speed of 0.05 m s^−1^, the wear rate of the solution-treated samples increased by 14% compared to the as-cast condition. However, the wear rate of the peak-aged samples was reduced by 18%, and the over-aged samples showed wear rates similar to the solution-treated ones.

Similarly, at the conditions of 20 N load and a sliding speed of 0.1 m s^−1^, the results showed a reduction in wear rates of 14% for the solution treatment and 35% for the peak-aged samples. The over-aged samples showed a wear rate equivalent to that of the as-cast samples.

The dispersion of data from the wear tests observed for the 20 N load and 0.1 m s^−1^ of sliding speed was much greater than for the other tests. The presence of a significant statistical deviation is typically indicative of instabilities on the wear process, as well as material transfer and loss from the samples, as shown below.

In general, wear increases with the applied load in most alloys, and it was observed for the ZX11 alloy for most of the conditions except for the as-cast samples. This suggests that the wear behavior of the ZX11 is influenced by its microstructure, which is particularly different in the as-cast condition, where the quasi-continuous network of precipitated phases at GBs provides a preferential path for crack propagation.

Additionally, the wear rates of the ZX11 alloy increases with sliding speed for the ZX11 alloy in most conditions, although wear is not affected much by the sliding speed in the solution-treated condition. In the case of the ZE41A alloy [27], wear rates have a double dependence on sliding speed; at very low speeds, the wear rate is higher due to the predominance of oxidative wear; it decreases at medium speeds; and then it increases again at higher speeds where abrasive wear becomes dominant.

Other alloys, such as the AZ91 and AZ91 modified with up to 3 wt.% of lanthanum-based rare earths, showed wear rates similar to those of the as-cast ZX11 alloy, but lower than the measured ones in the peak-aged condition [28]. However, they used a steel counterbody, which does not make possible a straightforward comparison of the results. In a similar system (AZ91 with 2 wt.% Y), but obtained by extrusion, they observed that the application of a T6 heat treatment resulted in a reduction in the wear rate that was greater for higher loads (15% reduction for 5 N loads and 35% reduction for 70 N) [29]. Nevertheless, they did not provide the results for the solution-treated alloy, so the effect of the intermediate stage was not shown. Also, a Mg-Zn-Y-Zr alloy showed lower wear rates in the initial state, and the application of heat treatments reduced them even more [22]. This observation is similar to what was observed in the ZX11 alloy, in which the as-cast samples showed a quasi-continuous phase at GBs and presented greater wear rates than the conditions with distributed precipitated particles.

Figure 5 shows the specific wear rate (*k*) for all the tested conditions, which is determined by dividing the wear rate (Figure 4) by the load used following Archard’s law (Equation (1)). In all cases, the specific wear rate was strongly dependent on the applied load. The specific wear at the load of 20 N was considerably lower across all conditions than for the 2 N condition, regardless of the sliding speed employed. The values observed ranged from 0.002 to 0.008 mm^3^ N^−1^ m^−1^ for all 2 N conditions, while for the 20 N load test, they were one order of magnitude lower, ranging from 0.0002 to 0.0008 mm^3^ N^−1^ m^−1^.

The *k* values were lower for the sliding speed of 0.05 m s^−1^ than for 0.1 m s^−1^ as is extracted from the comparison of Figure 4a,b. Therefore, the sliding speed has an impact on the specific wear rate. As observed, Archard’s law was not followed, since abrasive wear did not predominate. The condition with the lowest *k* value at 0.05 m s^−1^ was the peak-aged at both loads. At 0.1 m s^−1^ of sliding speed, the *k* value was minimal for the solution-treated condition at 2 N of load and the peak-aged one at 20 N of load. It is suggested that changes in the wear mechanisms take place.

The specific wear rate (*k*) is commonly used to assess the influence of load on the wear behavior and it is a sensitive factor to identify changes in the wear mechanisms. In the case of the ZX11 alloy, it has been observed that wear (Figure 4) is nearly unaffected by the load, causing the *k* values to reduce with the load (Figure 5). This behavior is different from what is typically observed in many systems [30,31]. There are two considerations to be made on this observation.

On the one hand, a Mg-based alloy has been used as the counterbody, i.e., the hardness of the two materials involved is similar. In most wear tests, a hard material is used as the counterbody, favoring the appearance of two-body abrasive wear [32]. However, this situation is not realistic within the human body, as no hard metals are present. The presence of similar materials implies that many other wear mechanisms could be present, as the evolution of *k* indicates.

On the other hand, the heat treatments applied have an impact on the wear rate of the ZX11 alloy. The presence of a quasi-continuous phase at the GBs in the as-cast condition caused fatigue wear due to crack propagation through the precipitated phases. This wear mechanism is not observed after the different heat treatments, due to the dissolution of the Ca_2_Mg_6_Zn_3_ phase at the GBs. Therefore, the evolution of the microstructure through the different heat treatment conditions changes the response of the alloy during its contact with the counterbody.

### 3.3. Friction Coefficient

The friction coefficients obtained for the ZX11 alloy samples were measured and recorded during testing, and the results are shown in Figure 6. In all tested samples, an unstable friction coefficient was observed for approximately 50 m of sliding distance. After this initial phase, the friction coefficient reached a relatively constant value.

In the steady state, the friction coefficients measured ranged between 0.27 and 0.46. It is important to indicate that the evolution of the friction coefficient with the heat treatments follows a similar trend to that of the wear rate.

The values observed for the friction coefficient are similar to those measured in wear tests where a magnesium alloy is tested against other metals in dry-sliding conditions. Ramesh et al. [33] obtained a similar average friction coefficient value for different Mg-Zn alloys applying 20 N and 2000 m of sliding distance using a pin-on-disc configuration with a steel disc as the counterbody. Zhao et al. [34] evaluated the friction coefficient of the as-cast and heat-treated Mg-6Gd-2Zn-0.4Zr (wt.%) alloy using a ball-on-disc configuration with stainless-steel balls, a sliding velocity of 20 mm s^−1^, a load of 20 N, and a 60 min test duration. Under these test conditions, they observed a very slight difference in the friction coefficient value between the as-cast and the heat-treated samples, ranging from 0.30 to 0.40 in all cases. Zhou et al. [35] showed in their work an average friction coefficient value of 0.3187 for an as-cast AZ91D alloy using a pin-on-disc configuration and steel balls, applying 10 N and 300 rpm for 15 min. Blau et al. [36] obtained friction coefficient values in the range of 0.30 to 0.35 for the AZ91D alloy manufactured by casting and thixomolding using three tribometers with different characteristics.

This research indicates that the combination of the ZX11 alloy with the AZ31 alloy produces friction levels similar to those observed between other Mg alloys and steel. Friction can arise either from abrasion or from the formation of micro-welds. In the case of the abrasive wear mechanism, one material penetrates into the other, and this mechanism is more frequent when a harder material comes into contact with a softer one [31]. In such cases, the friction is primarily limited by the shear resistance of the softer material, and the hardness of the harder material usually has limited impact. This mechanism causes the extraction of material from the samples and the formation of scratches and debris.

Another type of interaction between the testing material and the counterbody occurs when metallurgical unions are formed at the surface, and the ridges of both surfaces contact and weld. In this case, the upper limit of friction is associated with the lower shear resistance between the joint, the substrate, and the counterbody. This phenomenon causes material transference between the sample and the counterbody.

As a result, friction in the combination of the ZX11 alloy with the AZ31 alloy is limited by the shear resistance of both alloys. The evaluation of the composition at the surface of the samples can indicate if material transference is taking place, and the study of the surface morphology and of the debris formed during the wear test provides relevant information on the wear mechanism affecting the contact.

### 3.4. Determination of the Elementary Composition of the Surface

The surface composition of the ZX11 samples was examined using EDS analysis to assess any modifications caused by the wear tests. The evolution of the surface can come from the oxidation of the surface or from transference of material from the counterbody. Figure 7 represents the atomic percentage of oxygen detected on the tested sample surfaces. The results indicate that all the samples exhibited oxygen levels above 25%, but significant variations in the oxygen proportion were observed. In most conditions, a higher oxygen content was observed in tests conducted with higher loads, implying the formation of a greater quantity of oxides on the surface, which were also more stable.

Additionally, it was observed that at the low sliding velocity, the wear rates (Figure 4a) and oxygen content on the surface (Figure 7a) exhibited a proportional relationship. The friction coefficients of the ZX11 alloy at the low sliding velocity (Figure 6a) show an inverse relationship with the presence of oxygen (Figure 6a). This indicates that the oxides formed during the wear test have a lubricant effect during the wear test that resulted in a lower wear rate. Also, the as-cast sample shows a different behavior than the other systems, with a reduction in the amount of oxygen when the load applied increases, while the effect of load was lower for the samples in the peak-aged condition.

The results obtained at the highest speed (Figure 4b and Figure 7b) do not follow the same tendency. The amount of oxygen on the surface was lower and increased with the applied load (Figure 7b), but there is no clear correlation with the friction coefficients or the wear rate. This suggests that there are other wear mechanisms involved in the evolution of the surface of the samples during the wear tests.

Figure 8 shows the relationship between the wear rate and the friction coefficient with the atomic percentage of oxygen. In general, the increase in the wear rate corresponds to an increase in the O_2_ content found. In the case of the friction coefficient, the observed trend is the inverse. As the friction coefficient decreases, so does the atomic percentage of oxygen.

Aluminum was detected on the surfaces of the tested samples, even though the ZX11 alloy does not contain Al in its composition. However, the counterbody material, the AZ31 magnesium alloy, contains a small proportion of Al. Hence, the presence of aluminum on the surface of the tested samples indicates material transference from the counterbody to the ZX11 samples. The quantity of aluminum observed on each specimen is shown in Figure 9, with values exceeding 1% of aluminum content. Considering that the counterbody contains 3 wt.% of aluminum, an aluminum proportion of 1% suggests that approximately one third of the sample surface is covered by material transferred from the counterbody.

For applied loads of 2 N in the as-cast and solution-treated ZX11 alloy (Figure 9a,b), as well as for the 20 N in the as-cast 0.05 m s^−1^ and solution-treated 0.1 m s^−1^ conditions, the amount of content remained below 0.25%. However, in the remaining conditions, the aluminum content ranged from 0.50 to 1.00%. In general, there is a proportional relationship between the presence of aluminum and oxygen, indicating the possibility of a similar mechanism contributing to their appearance.

### 3.5. Wear Mechanisms

To explain the differences observed in the wear processes of the tested samples, an analysis of the wear mechanisms present in each condition has been conducted. To evaluate the predominant mechanisms, both the surface of the samples and the debris formed have been studied. Figure 10 shows the worn surfaces of the as-cast ZX11 alloy pins under the various test conditions applied. In all conditions and samples, fine grooves parallel to the sliding direction were visible, indicating the occurrence of abrasion (red arrow). Additionally, the presence of oxygen in all sample surfaces indicates the occurrence of oxidation wear (green arrows). Oxidation wear is common in magnesium alloys due to its high reactivity and is characterized by the formation of spots or platelets on the magnesium alloy surface. However, the observed behavior exhibits distinct characteristics among the different samples studied.

Figure 10a–d show two types of oxides with different contrast. The darker oxide is enriched in aluminum, indicating the presence of material transferred from the AZ31 counterbody to the pin surface. This oxide is more visible in Figure 10a–c because it does not cover the entire surface. However, its proportion is significantly higher in the 20 N–0.1 m s^−1^ condition (Figure 10d), where it completely covers the surface.

At the lower speed and with an increase in applied load, in addition to abrasion and oxidative wear, deformation layers of the alloy become noticeable, and delaminated zones were observed (inset in Figure 10b—blue arrow). At higher speeds, a more continuous surface is observed, indicating the occurrence of plastic deformation (yellow arrow) [17,18]. The 2 N load-tested sample (Figure 10c) showed more oxides than the 20 N tested one (Figure 10d).

Figure 11 presents evidence of the main wear mechanisms observed in the ZX11 solution-treated samples under various test conditions. Abrasion grooves and the presence of oxygen were observed on all worn surfaces. Additionally, Figure 11a reveals cracks perpendicular to the sliding direction, indicating the occurrence of delamination wear. In this wear mechanism, subsurface cracks propagate and extend until they reach the surface as cracks that are perpendicular to the sliding direction.

An increase in applied load (Figure 11b,d) resulted in rougher surfaces, which correspond to higher wear rates, as illustrated in Figure 3, and a greater amount of material transferred from the counterbody (Figure 9a,b). A more homogenous surface was observable in the 2 N–0.1 ms^−1^ condition (Figure 11c). No signs of plastic deformation were observed in the solution-treated alloy, unlike the as-cast alloy.

Figure 12 shows that the peak-aged ZX11 alloy samples tested under various load and sliding velocity conditions exhibit a rough surface because of the presence of abrasion and oxidation, with oxidation being the dominant mechanism. These samples also exhibited the highest hardness. High contents of O and Al at the surface (Figure 7 and Figure 9) indicate oxidation and material transfer from the counterbody to the pin surface.

According to several studies [17,37,38,39], a mechanically mixed layer (MML) of oxides forms at the surface, incorporating material from the counterbody. These oxide layers possess high hardness values and remain stable within certain ranges of load and sliding velocity. If the oxide layer formed is detached during the test, the wear rate increases, as a new wear mechanism appears that favors the material removal from the sample surface. However, the presence of a stable MML layer reduces both the wear rate and the friction coefficient. Under more energetic conditions, deformation of the material at the surface is observed, which is attributable to the high temperatures reached in the contact zones between the materials.

Figure 13 shows the worn surfaces of the ZX11 alloy in the over-aged condition. Oxidation wear was also the predominant mechanism, along with abrasion. The wear rates of the over-aged samples (Figure 4a,b) can be attributed to their hardness, as explained for the peak-aged ZX11 alloy samples. The over-aged samples present a hardness value of about 69 HV, which is higher than the hardness of the counterbody employed in the wear tests (64 ± 3 HV). This difference in hardness likely contributes to the observed wear rates.

Figure 14 shows the debris generated in the wear tests of the as-cast ZX11 alloy. The debris found presents two different morphologies with differentiated compositions (marked with the letters A and B in Figure 14). The EDS mapping in Figure 14 reveals that the debris marked as A exhibits a higher Al content, indicating that it comes from the AZ31 alloy counterbody. These debris particles are formed through abrasion from the counterbody during the wear test, resulting in an elongated, curved, or spiral morphology. On the other hand, debris marked as B has a higher O content and is associated with the oxide layer formed during the wear test. These oxidized particles initially have a small size and irregular morphology, and agglomerate due to repeated sliding, forming larger particles and even a surface layer with some protective properties against wear. These two types of debris were observed throughout the different heat-treated tested samples.

Figure 15 and Figure 16 provide information on the O and Al (atomic %) content of the debris formed in each test under the different load and sliding velocity conditions. The results suggest that at lower loads and sliding velocities, the oxidative phenomenon dominates, as was observed in other alloys such as AM50B and AZ91 [17,18].

Figure 16 indicates that the analyzed debris from all samples presents a similar Al content. This indicates that material from the AZ31 alloy used as the counterbody was removed in all test conditions for all the heat-treated ZX11 samples.

In general, the microstructural changes of the ZX11 alloy after the heat treatments applied do not always result in improved wear resistance. However, the heat treatments prevent the plastic deformation wear mechanism in the ZX11 magnesium alloy. Shanthi et al. [39] found that grain refinement of the recycled AZ91 alloy significantly improved the mechanical properties of the alloy but did not necessarily improve its dry sliding wear resistance. The increased wear rate of the new samples, compared to the as-cast AZ91 alloy, was attributed to the brittleness, which promotes the abrasion mechanism. The presence of an MML oxide layer and its protective nature level of protection did not seem to be correlated with the grain size or the pin properties.

Additionally, the presence of a quasi-continuous phase at grain boundaries in the studied ZX11 alloy appears to facilitate wear evolution through the crack propagation within the material. This morphology, which is interesting for the control of corrosion in chloride media [16], does not inhibit the formation of oxides at the surface. Consequently, both mechanical and oxidative wear mechanisms are favored in the presence of the as-cast morphology. The breakdown of the continuous morphology of precipitated phases at the GBs into discrete precipitates reduces the effect of fatigue wear mechanisms, while the increase in hardness reduces abrasion. Therefore, the peak-aged condition exhibits the best wear behavior among the tested conditions.

## 4. Conclusions

The following conclusions can be drawn:-The microstructure of the ZX11 alloy varied with heat treatments. The as-cast condition exhibited a continuous precipitated phase at the grain boundaries. After the heat treatments, the precipitates appeared within the grains, and the size of the precipitated phases increased with longer aging time.-The hardness of the ZX11 alloy varied with the heat treatment. The as-cast conditions exhibited a hardness of 62 HV. After heat treatments, the hardness increased to 71 HV at the peak-aged condition, and then decreased to 69 HV after aging for 24 h.-The wear rates and the friction coefficients were not significantly affected by the applied load. However, there was a direct relationship between the friction coefficient and the wear rate, showing an inverse relationship between hardness and wear rate.-Abrasion and oxidation were identified as the main wear mechanisms observed in the ZX11 alloy under different wear conditions. Delamination was also observed in most tested systems. Plastic deformation was only observed in the as-cast ZX11 alloy.-The wear debris formed during the tests consisted of material from the ZX11 alloy and the AZ31 counterbody. The first one shows an irregular morphology and it has a higher O content, being related to the oxide layer formed during the wear test. The second one is more elongated and exhibits a higher Al content, which corresponds to that coming from the AZ31 alloy counterbody. Material transfer from the counterbody was particularly relevant in the heat-treated conditions.-The presence of a quasi-continuous phase at grain boundaries in the as-cast condition led to fatigue wear due to the propagation of cracks through the precipitated phases, while oxidative wear was also present. This study allows for identifying the effect that the change in the microstructure causes in the wear behavior of the ZX11 alloy.

## Figures and Tables

**Figure 1 materials-17-00070-f001:**
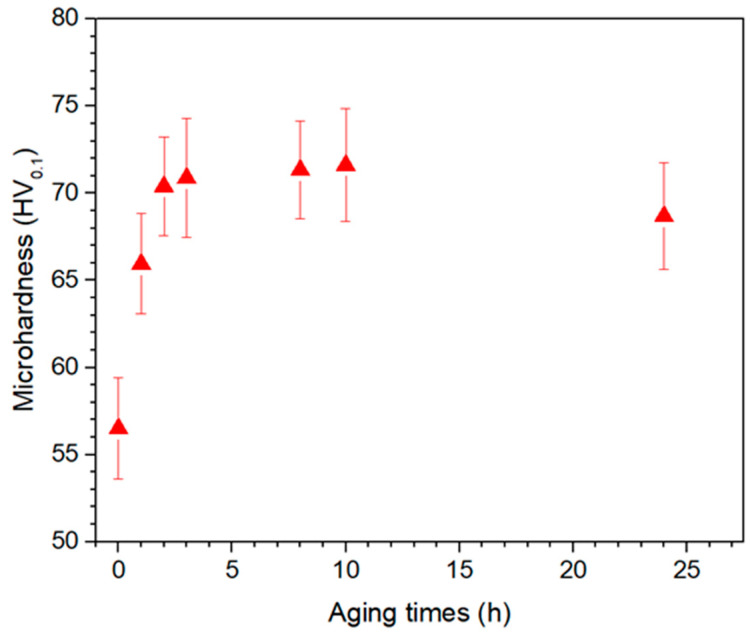
Microhardness evolution with aging time at 180 °C for the ZX11 alloy.

**Figure 2 materials-17-00070-f002:**
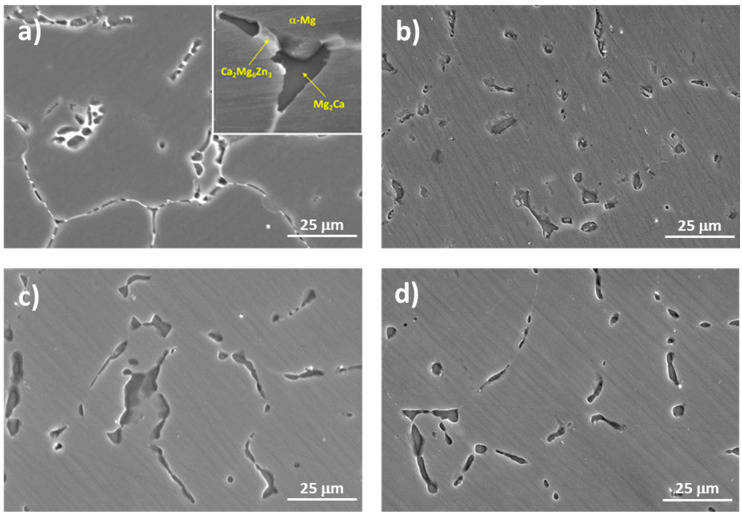
SEM micrographs of the ZX11 alloy samples: (**a**) as-cast, (**b**) solution-treated, (**c**) peak-aged, and (**d**) over-aged.

**Figure 3 materials-17-00070-f003:**
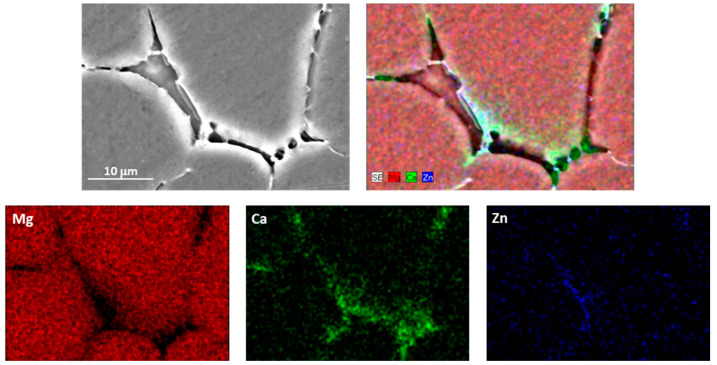
Detail of the precipitated phases in a grain boundary of the as-cast ZX11 alloy.

**Figure 4 materials-17-00070-f004:**
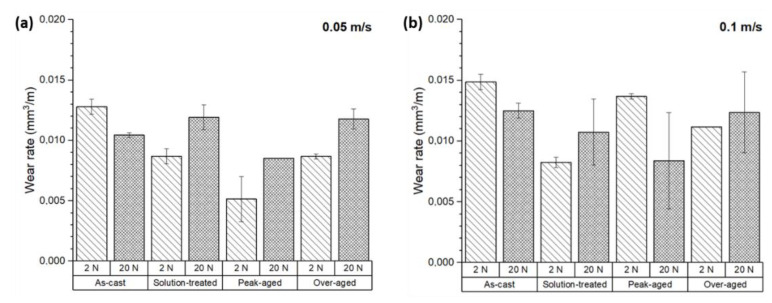
Wear rate of the ZX11 alloy under the following conditions of load and sliding speed: (**a**) 2 N and 20 N at 0.05 m s^−1^; (**b**) 2 N and 20 N at 0.1 m s^−1^.

**Figure 5 materials-17-00070-f005:**
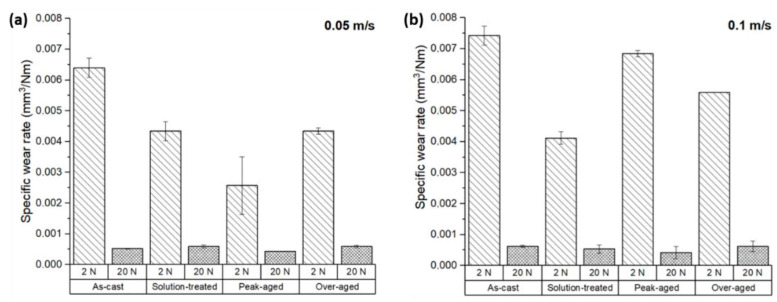
Specific wear rates for the ZX11 alloy: (**a**) 2 N and 20 N at 0.05 m s^−1^; (**b**) 2 N and 20 N at 0.1 m s^−1^.

**Figure 6 materials-17-00070-f006:**
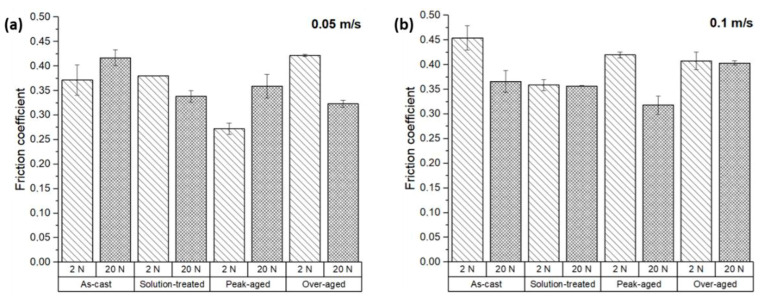
Friction coefficients of the ZX11 alloy: (**a**) 2 N and 20 N at 0.05 m s^−1^; (**b**) 2 N and 20 N at 0.1 m s^−1^.

**Figure 7 materials-17-00070-f007:**
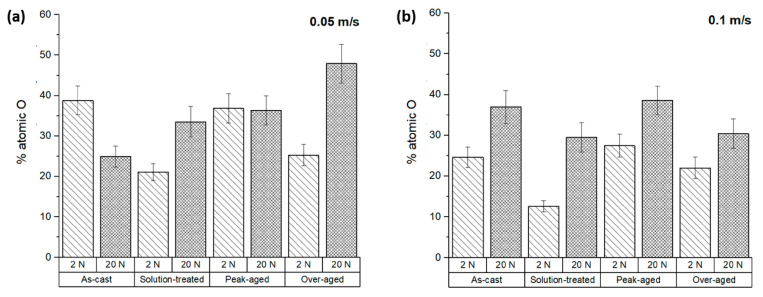
Atomic percentage of oxygen on the worn surfaces of the ZX11 alloy**:** (**a**) 2 N and 20 N at 0.05 m s^−1^; (**b**) 2 N and 20 N at 0.1 m s^−1^.

**Figure 8 materials-17-00070-f008:**
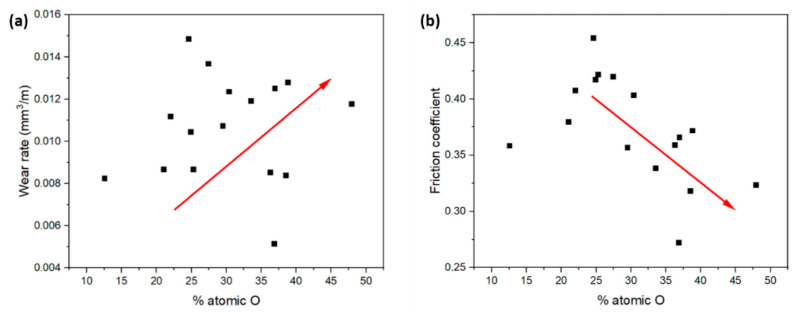
Relationship between (**a**) wear rate and atomic percentage of oxygen; (**b**) friction coefficient and atomic percentage of oxygen.

**Figure 9 materials-17-00070-f009:**
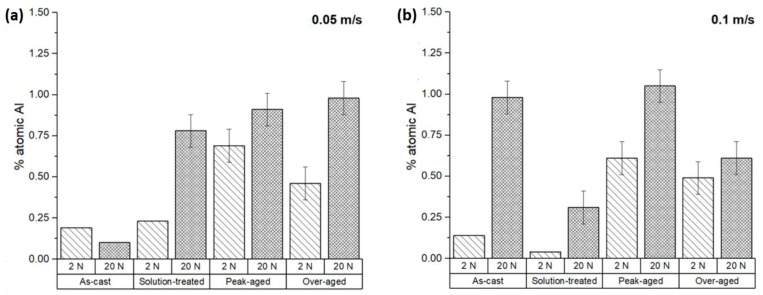
Atomic percentage of aluminum on the worn surface of the ZX11 alloy: (**a**) 2 N and 20 N at 0.05 m s^−1^; (**b**) 2 N and 20 N at 0.1 m s^−1^.

**Figure 10 materials-17-00070-f010:**
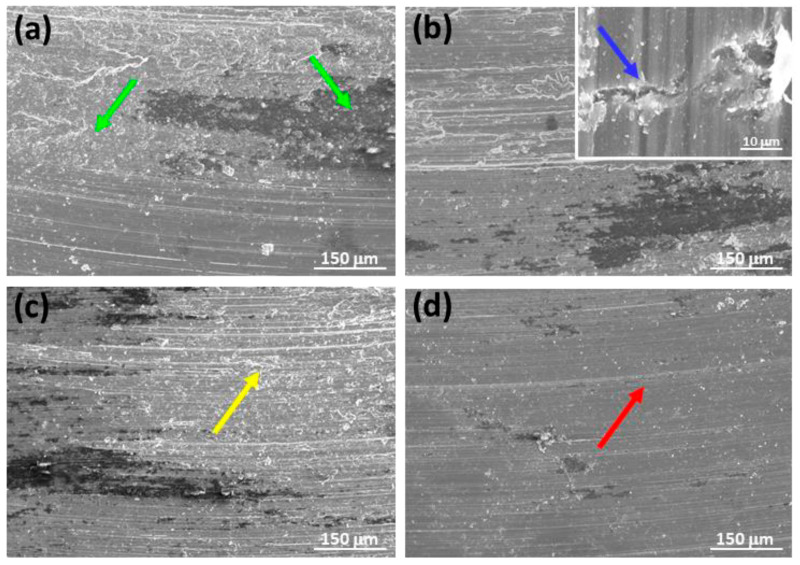
Worn tracks observed for the as-cast ZX11 alloy at the different wear conditions: (**a**) 2 N, 0.05 m s^−1^; (**b**) 20 N, 0.05 m s^−1^; (**c**) 2 N, 0.1 m s^−1^; (**d**) 20 N, 0.1 m s^−1^.

**Figure 11 materials-17-00070-f011:**
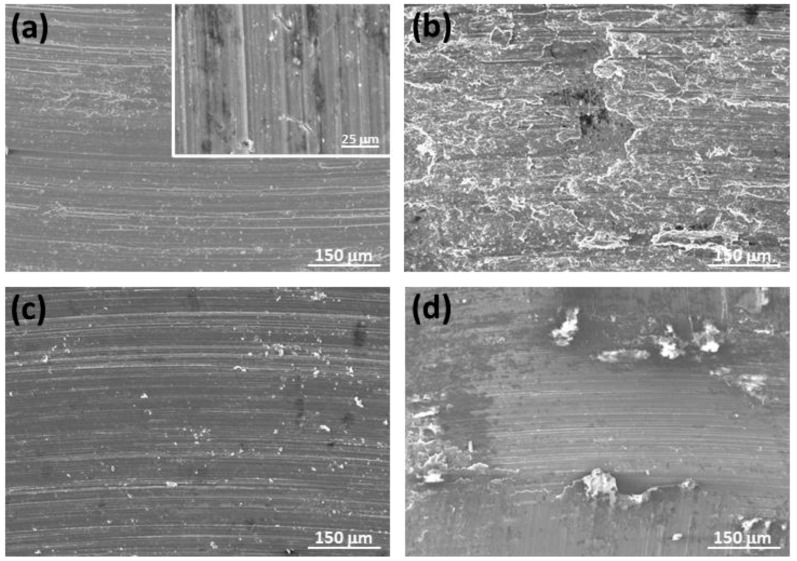
Worn tracks observed for the solution-treated ZX11 alloy at the different wear conditions: (**a**) 2 N, 0.05 m s^−1^; (**b**) 20 N, 0.05 m s^−1^; (**c**) 2 N, 0.1 m s^−1^; (**d**) 20 N, 0.1 m s^−1^.

**Figure 12 materials-17-00070-f012:**
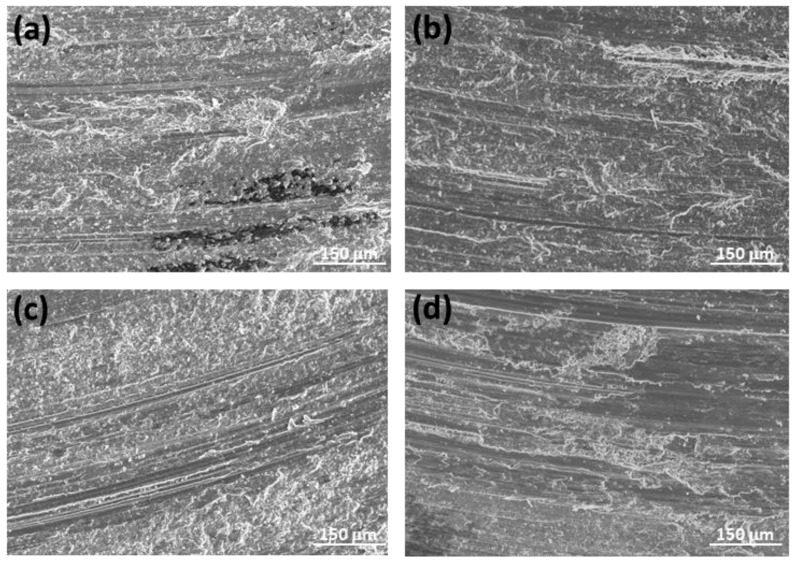
Worn tracks observed for the peak-aged ZX11 alloy at the different wear conditions: (**a**) 2 N, 0.05 m s^−1^; (**b**) 20 N, 0.05 m s^−1^; (**c**) 2 N, 0.1 m s^−1^; (**d**) 20 N, 0.1 m s^−1^.

**Figure 13 materials-17-00070-f013:**
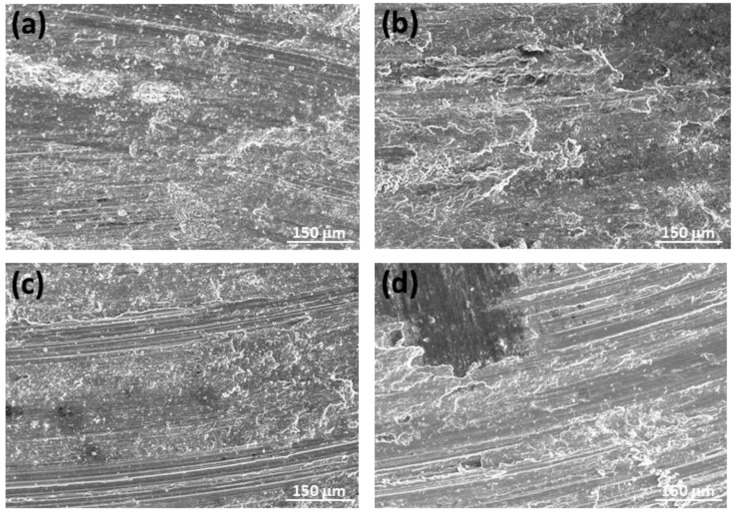
Worn tracks observed for the over-aged Mg-1Zn-1Ca alloy at the different wear conditions: (**a**) 2 N, 0.05 m s^−1^; (**b**) 20 N, 0.05 m s^−1^; (**c**) 2 N, 0.1 m s^−1^; (**d**) 20 N, 0.1 m s^−1^.

**Figure 14 materials-17-00070-f014:**
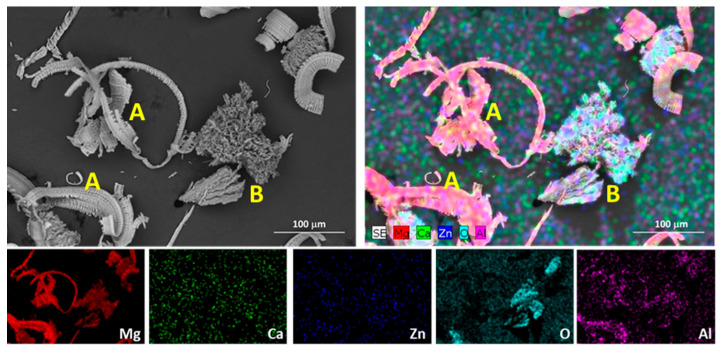
SEM micrograph and EDS mapping of composition of the wear debris found in the as-cast ZX11 alloy samples at 2 N and 0.1 m s^−1^. A and B represent the different debris found after wear tests.

**Figure 15 materials-17-00070-f015:**
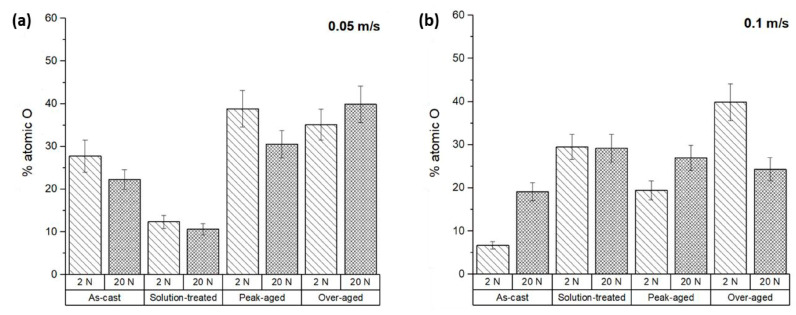
Atomic percentage of oxygen on the debris of the samples tested. ZX11 alloy under the following conditions of load and sliding speed: (**a**) 2 N and 20 N at 0.05 m s^−1^; (**b**) 2 N and 20 N at 0.1 m s^−1^.

**Figure 16 materials-17-00070-f016:**
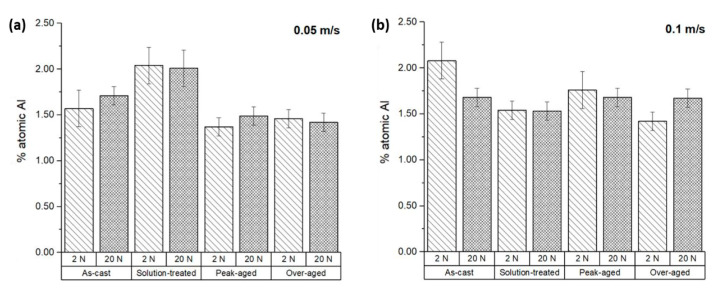
Atomic percentage of aluminum on the debris. ZX11 alloy under the following conditions of load and sliding speed: (**a**) 2 N and 20 N at 0.05 m s^−1^; (**b**) 2 N and 20 N at 0.1 m s^−1^.

**Table 1 materials-17-00070-t001:** Composition in wt.% of the ZX11 magnesium alloy.

Ca	Zn	Fe	Cu	Ni	Mg
1.47	1.01	0.0071	0.0015	0.0007	Bal.

**Table 2 materials-17-00070-t002:** Summary of heat treatments, aging times, and the Vickers microhardness of the samples selected for wear testing.

Alloy	Sample	Aging Time (h)	HV0.1
ZX11	As-cast	-	62 ± 6
	Solution-treated	0	56 ± 3
	Peak-aged	3	71 ± 3
	Over-aged	24	69 ± 3

**Table 3 materials-17-00070-t003:** Composition in wt.% of the AZ31 alloy used as counterbody.

	Al	Ca	Cu	Fe	Mn	Ni	Si	Zn	Zr	TO *	Mg
AZ31	2.9	<0.005	<0.0005	0.005	0.17	0.0005	<0.005	0.96	<0.005	<0.3	Bal.

TO *: Total Others.

## Data Availability

Data are contained within the article.

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
