# Peer review of "Microstructure and Wear Behavior of Heat-Treated Mg-1Zn-1Ca Alloy for Biomedical Applications"

_materials, 2023, doi:10.3390/ma17010070_

Round 1

Reviewer 1 Report

Comments and Suggestions for Authors

In the present work, the effect of heat treatment on the sliding wear behaviors of Mg-1Zn-1Ca alloy were analyzed. The quality of this manuscript is suitable for publication in this journal. I suggest that the manuscript can be published with minor revisions.

1. What is the difference between actual service conditions and experimental conditions of biodegradable Mg-1Zn-1Ca alloy?

2. How to evaluate the mechanical and physical properties of biodegradable magnesium alloys under service conditions by studying their sliding wear behavior? What is the internal connection?

Comments on the Quality of English Language

Good!

Author Response

  1. What is the difference between actual service conditions and experimental conditions of biodegradable Mg-1Zn-1Ca alloy?

Under service conditions the biodegradable Mg-1Zn-1Ca alloy is in contact with fluids and it works at body temperature. Once the dry wear behavior of this alloy has been established, the next step would consist of testing them in immersion in a simulated body fluid and at temperature.

  1. How to evaluate the mechanical and physical properties of biodegradable magnesium alloys under service conditions by studying their sliding wear behavior? What is the internal connection?

There are two main points in the connection between wear testing and biodegradation of magnesium alloys. The materials used as prostheses for traumatological healing are usually in contact with other materials, either natural bones or other parts of the prostheses.

In most cases, there is not sliding between the prostheses and the bone, but in others there may be. There can be sliding between the Mg prosthesis and the bone in some types of implants. Also, in the case of biodegradable alloys as their size changes. This can drive to local sliding of the different parts of the prosthesis system that could derive in wear of any of both parts. In this case, the knowledge and control of wear is of relevance.

We acknowledge the reviewer for the improvement of our work.

Reviewer 2 Report

Comments and Suggestions for Authors

The authors conducted a study titled 'Microstructure and Wear Behavior of Heat-Treated Mg-1Zn-1Ca Alloy for Biomedical Applications'.

Several major issues need to be addressed and resolved:

1. The background study given in the introduction is insufficient, hence the motivation and problem statement of the study is not clear.

2. Some important supporting results that are necessary e.g. XRD, EDX, morphology of intermetallics/secondary phases are not included, hence it is difficult to correlate the effect of heat treatments on both microstructures and wear properties.

3. Most of the results are given with basic/rudimentary analysis, without any in-depth discussion and correlation with the results from the present study and other relevant studies from available literature.

Details of comments/suggestions can be ound in the PDF attached.

Comments on the Quality of English Language

English is mostly okay, just please double check and revise on the inconsistent fonts throughout the manuscript. Also, some words/phrases used to describe certain aspects are not suitable and do not precisely reflect what they are meant to confer. Refer to the comments in the attached PDF.

Author Response

Several major issues need to be addressed and resolved:

  1. The background study given in the introduction is insufficient, hence the motivation and problem statement of the study is not clear.
  2. Some important supporting results that are necessary e.g. XRD, EDX, morphology of intermetallics/secondary phases are not included, hence it is difficult to correlate the effect of heat treatments on both microstructures and wear properties.
  3. Most of the results are given with basic/rudimentary analysis, without any in-depth discussion and correlation with the results from the present study and other relevant studies from available literature.

Details of comments/suggestions can be found in the PDF attached.

English is mostly okay, just please double check and revise on the inconsistent fonts throughout the manuscript. Also, some words/phrases used to describe certain aspects are not suitable and do not precisely reflect what they are meant to confer. Refer to the comments in the attached PDF.

The document has been revised, the discussions have been improved and all the consideration in the pdf have been fulfilled.

We acknowledge the reviewer for the detailed analysis made to improve our work.

Reviewer 3 Report

Comments and Suggestions for Authors

Review of Microstructure and Wear Behavior of Heat-Treated Mg-1Zn-1Ca Alloy for Biomedical Applications

First of all, this research paper has investigated the microstructure and wear behavior of Mg alloy (with different heat treated Mg specimen) for biomedical applications. The main focus of this paper is investigating the solution-treated, peak-aged, and over-aged of Mg-1Zn-1Ca magnesium alloy. There are some questions and recommendations as follows:

1) In Table 1, there is a list of chemical composition. Did you perform any spectral analysis, or did you get it from producer?

2) In wear tests, it will be good for readers if you add simple wear test configuration drawing

3) The counter body is an AZ31 magnesium alloy plate, you gave the chemical composition but the hardness of AZ31 is missing. It is very important during the pin-on-disc wear test.

4) During the wear test, all friction coefficients obtained for the ZX11 alloy samples are measured and recorded during testing, but you gave only the mean friction coefficients. I think it will be good for readers, if you add distance vs COF plot. During the tests, there will be some fluctuation, increase and decrease in plot. According to these fluctuations you can comment about wear mechanism. There is a work which you can access in this link (https://journals.pan.pl/Content/117806/PDF/AMM-2021-1-26-Urtekin.pdf)
DOI: 10.24425/amm.2021.134777Investigation of Tribological and Mechanical Properties of Biodegradable AZ91 Alloy Produced by Cold Chamber High Pressure Casting MethodArchives of Metallurgy and Materials, 2021, vol. 66, No:1, 205-216. It will be good if you cite this work. It is similar work which investigates the AZ91 Mg alloy under dry wear pin-on-disc test.

5) Figure 4, Figure 7, Figure 8, Figure 14, and Figure 15 has lower quality in comparison to other figures. Please update these figures.

6) In Figure 5, for peak aged specimen under 2 N load, wear test which is performed under 0.1 m/s has higher specific wear rate than one performed under 0.05 m/s. Can you say something about why it happens

7) Rest of analysis such as SEM, EDS and especially precipitated phases in a grain boundary of the as-cast ZX11 alloy are very logical and well designed. My last recommendation is there are some writing errors. Please read the work from abstract to conclusion.

Comments on the Quality of English Language

No Comments 

Author Response

First of all, this research paper has investigated the microstructure and wear behavior of Mg alloy (with different heat treated Mg specimen) for biomedical applications. The main focus of this paper is investigating the solution-treated, peak-aged, and over-aged of Mg-1Zn-1Ca magnesium alloy. There are some questions and recommendations as follows:  

  1. In Table 1, there is a list of chemical composition. Did you perform any spectral analysis, or did you get it from producer?  

We get the information from producer. 

  1. In wear tests, it will be good for readers if you add simple wear test configuration drawing. 

A simple wear test configuration drawing has been added. 

  1. The counter body is an AZ31 magnesium alloy plate, you gave the chemical composition but the hardness of AZ31 is missing. It is very important during the pin-on-disc wear test.  

The AZ31 hardness value has been added in the Experimental procedure. 

  1. During the wear test, all friction coefficients obtained for the ZX11 alloy samples are measured and recorded during testing, but you gave only the mean friction coefficients. I think it will be good for readers, if you add distance vs COF plot. During the tests, there will be some fluctuation, increase and decrease in plot. According to these fluctuations you can comment about wear mechanism. There is a work which you can access in this link (https://journals.pan.pl/Content/117806/PDF/AMM[1]2021-1-26-Urtekin.pdf) DOI: 10.24425/amm.2021.134777, Investigation of Tribological and Mechanical Properties of Biodegradable AZ91 Alloy Produced by Cold Chamber High Pressure Casting Method, Archives of Metallurgy and Materials, 2021, vol. 66, No:1, 205-216. It will be good if you cite this work. It is similar work which investigates the AZ91 Mg alloy under dry wear pin-on-disc test.  

We appreciate your suggestion. In this research, we focus on analyzing the effect that the sliding speed and the applied load have on different wear parameters. For this reason, we have decided to represent all graphs in the same way. However, no significant changes were observed in the friction coefficient values with distance after 50 m. 

The information extracted from this research has been included in the text. 

  1. Figure 4, Figure 7, Figure 8, Figure 14, and Figure 15 has lower quality in comparison to other figures. Please update these figures.  

The images have been improved. 

  1. In Figure 5, for peak aged specimen under 2 N load, wear test which is performed under 0.1 m/s has higher specific wear rate than one performed under 0.05 m/s. Can you say something about why it happens?  

Additional information has been added to the manuscript. The wear rate is most affected by the sliding speed than applied load, causing that the specific wear rate reduces with load. It could be suggested that a MML layer of different stability appears in both samples tested at different sliding speed. 

7) Rest of analysis such as SEM, EDS and especially precipitated phases in a grain boundary of the as-cast ZX11 alloy are very logical and well designed. My last recommendation is there are some writing errors. Please read the work from abstract to conclusion. 

We have revised the writing throughout the entire manuscript. 

Reviewer 4 Report

Comments and Suggestions for Authors

Title: Microstructure and Wear Behavior of Heat-Treated Mg-1Zn-1Ca Alloy for Biomedical Applications

This document delves into the microstructure and wear properties of an Mg-1Zn-1Ca alloy subjected to various heat treatments. Specifically, it highlights that the as-cast sample exhibited a continuous intermetallic phase at the grain boundaries, whereas the heat-treated samples revealed discrete precipitated particles within the grains. The wear behavior was systematically assessed using a pin-on-disc configuration, unveiling distinct wear mechanisms under different test conditions. While the authors have conducted commendable work, a response to the following query is requisite.

  1. Casting Process and Molten Magnesium Protection:
    • Which casting process is employed in the current study?
    • How is the molten magnesium protected during the casting process?
  2. Heat Treatment Processes for Mg-Alloy:
    • What are the various heat treatment processes applicable to Mg-alloys, and could the authors introduce them?
    • What criteria were used to select the specific heat treatment processes applied in the current study?
  3. ASTM Standard for Wear Test:
    • Which ASTM standard for wear testing is employed in the study?
  4. Choice of Rectangular Wear Test Sample:
    • Authors typically use circular wear test samples. Why did the authors opt for rectangular wear test samples?
  5. Counterpart Material and Wear Debris Quantification:
    • Considering the counterpart is also a magnesium alloy, how do the authors plan to address the potential wear of both pin and disc, and the challenge of quantifying wear debris?
  6. Limited Load Range in Wear Test:
    • Why was the wear test conducted under only two loads (2N and 20N)? How might the wear behavior vary between these loads?
  7. X-ray Diffraction (XRD) Data:
    • If X-ray diffraction was performed for phase identification, is it possible to include the XRD data in the manuscript?
  8. Possible Wear Mechanisms in Magnesium Alloy:
    • What are the different types of wear mechanisms expected in magnesium alloy?
    • Which wear mechanisms are prevalent under 2N and 20N loads?
  9. Effect of Load and Sliding Speed on Counterpart Wear:
    • How does the load and sliding speed influence the wear of the counterpart material?
  10. Calculation of Specific Wear Rate:
    • Could you explain how the specific wear rate was calculated in the study?
  11. Calculation of Coefficient of Friction:
    • What methodology was employed to calculate the coefficient of friction?
  12. Elemental Analysis Prediction with EDS:
    • Is EDS analysis alone sufficient to accurately predict elemental composition?
  13. Identification of Wear Mechanism in SEM Images:
    • Since it is challenging to identify a single wear mechanism, can the authors highlight specific wear mechanisms on different SEM images?
  14. Differences in Wear Mechanisms Between As-Cast and Heat-Treated Samples:
    • How did the wear mechanisms differ between the as-cast and heat-treated samples?
  15. Variation in Wear Properties Under Different Test Conditions:
    • Were there any notable differences in the wear properties of the Mg-1Zn-1Ca alloy under different test conditions?
  16. Inclusion of Wear Debris Outcome in Conclusion:
    • The conclusion does not mention the outcomes from the wear debris (presented in Fig 13). Can this be addressed?
  17. Enhancement of Conclusion:
    • The conclusion could be improved by specifying the key outputs or findings of the manuscript.
  18. Inclusion of Additional Relevant Papers:
    • It would be beneficial if the authors could include references to papers related to the processing and wear mechanisms of magnesium alloys.

https://doi.org/10.1007/s40962-021-00747-9

https://doi.org/10.3390/ma14040990

https://doi.org/10.1016/j.jma.2018.05.006

Comments on the Quality of English Language

Minor English correction required.

Author Response

  1. Casting Process and Molten Magnesium Protection:
    • Which casting process is employed in the current study?
    • How is the molten magnesium protected during the casting process?

The alloy was supplied by Helmholtz Zentrum Geesthacht who performed the entire manufacturing process. They made it using the permanent mold indirect chill casting under a protective atmosphere of argon with 0.2% SF6 and held at 720 °C. This information can be found in the text and the corresponding literature.

  1. Heat Treatment Processes for Mg-Alloy:
    • What are the various heat treatment processes applicable to Mg-alloys, and could the authors introduce them?

The designation of the different heat treatments applicable to Mg and its alloys is shown in Table 1.

Table 1. Basic temperature designations [1].

[1] M. M. Avedesian, H. Baker, ASM Specialty Handbook: Magnesium and Magnesium Alloys, 1999, ASM International. ISBN: 978-0-87170-657-7

In this research, we carried out a T6 heat treatment.

    • What criteria were used to select the specific heat treatment processes applied in the current study?

A deep bibliographic search and the results obtained in a previous published manuscript [2] were used to define the manufacturing conditions chosen.

In addition to the as-cast condition, and the solution-treated condition, the samples were heat treated to reach the maximum hardness, which is the condition at which mechanical properties presumably achieve the peak, and an overaged condition that would provide greater ductility.

[2] N. Pulido-González, P. Hidalgo-Manrique, S. García-Rodríguez, B. Torres, J. Rams, Effect of heat treatment on the mechanical and biocorrosion behaviour of two Mg-Zn-Ca alloys, J. Magnes. Alloy. (2021). https://doi.org/10.1016/j.jma.2021.06.022.

  1. ASTM Standard for Wear Test:
    • Which ASTM standard for wear testing is employed in the study?

The wear test followed the ASTM G99-17 standard, although in the system used the pin was made of the material to be tested, i.e., the ZX11 Mg alloy, and the disc was a reference material, which was the AZ31 Mg alloy. Therefore, the test carried out does not fulfill the conditions that were used in the interlaboratory study that is referred in the ASTM G99-17 standard.

  1. Choice of Rectangular Wear Test Sample:
    • Authors typically use circular wear test samples. Why did the authors opt for rectangular wear test samples?

This type of morphology was used due to availability. Morphology does not affect the type of test performed as the imprint was a circle. In this case, the use of a laminated reference surface (disk/square sample) simplified the use of squared samples.

  1. Counterpart Material and Wear Debris Quantification:
    • Considering the counterpart is also a magnesium alloy, how do the authors plan to address the potential wear of both pin and disc, and the challenge of quantifying wear debris?

In this study, the composition of the two alloys is different. The most defining element is Al as the AZ31 contains a 3% Al, while the ZX11 does not have Al. Therefore, from the percentage of Al in the debris or in the transferred material it is possible to determine the proportion of material that comes from each material.

  1. Limited Load Range in Wear Test:
    • Why was the wear test conducted under only two loads (2N and 20N)? How might the wear behavior vary between these loads?

We have tested two extreme conditions to find the most significant differences in the wear behavior of this alloy. Between these loads, the wear behavior would be intermediate between those tested. The differences found indicate that there is no strong change in mechanisms within the two loads, suggesting the validity of the conditions considered. For a later study it would be interesting to expand the loads and speeds used.

  1. X-ray Diffraction (XRD) Data:
    • If X-ray diffraction was performed for phase identification, is it possible to include the XRD data in the manuscript?

This information is included in a previous published manuscript [3].

[3] N. Pulido-González, B. Torres, S. García-Rodríguez, P. Rodrigo, V. Bonache, P. Hidalgo-Manrique, M. Mohedano, J. Rams, Mg–1Zn–1Ca alloy for biomedical applications. Influence of the secondary phases on the mechanical and corrosion behaviour, J. Alloys Compd. 831 (2020) 1–15. https://doi.org/10.1016/j.jallcom.2020.154735.

New information has been added to the text, as well as the previously cited reference.

  1. Possible Wear Mechanisms in Magnesium Alloy:
    • What are the different types of wear mechanisms expected in magnesium alloy?

The expected wear mechanisms correspond to those observed in the present study: abrasive, oxidative, plastic deformation, delamination.

This information has been included with more detail in the text.

    • Which wear mechanisms are prevalent under 2N and 20N loads?

No exclusive wear mechanisms have been observed for each of the loads used. However, the presence of oxidation was more pronounced in the samples tested at 20N, as shown in the manuscript.

  1. Effect of Load and Sliding Speed on Counterpart Wear:
    • How does the load and sliding speed influence the wear of the counterpart material?

In this study, we could evaluate it based on the amount of aluminum found on the worn surfaces. There is a material transfer from the counterbody to the tested samples. Moreover, the Al content of the debris formed after each test has been analyzed.

  1. Calculation of Specific Wear Rate:
    • Could you explain how the specific wear rate was calculated in the study?

The specific wear was calculated form the coefficient of the wear rate divided by the load used, i.e., the wear volume (V) per sliding distance (L) and load (W).

This information can be found in the text.

  1. Calculation of Coefficient of Friction:
    • What methodology was employed to calculate the coefficient of friction?

The device has a strain gage that measures the transversal load on the pin. This force corresponds to the friction force and divided by the load applied provides the friction coefficient.

  1. Elemental Analysis Prediction with EDS:
    • Is EDS analysis alone sufficient to accurately predict elemental composition?

Although EDS is a semiquantitative technique, it has result to be a useful tool to compare different samples at a compositional level. It is especially interesting to distinguish between the counterbody and the sample, as they have different composition.

This consideration has been included in the text.

  1. Identification of Wear Mechanism in SEM Images:
    • Since it is challenging to identify a single wear mechanism, can the authors highlight specific wear mechanisms on different SEM images?

The information on the SEM images requested by the reviewer has been added to the text.

  1. Differences in Wear Mechanisms Between As-Cast and Heat-Treated Samples:
    • How did the wear mechanisms differ between the as-cast and heat-treated samples?

The main difference found is that the presence of a quasi-continuous phase at the grain boundaries in the as-cast condition caused fatigue wear due to crack propagation through the precipitated phases, while oxidative wear was also present.

This constitutes one of the most relevant conclusions of the work.

  1. Variation in Wear Properties Under Different Test Conditions:
    • Were there any notable differences in the wear properties of the Mg-1Zn-1Ca alloy under different test conditions?

The wear rates can be reduced by 25% and 50 % from the values measured in the as-cast sample with different heat treatments. This implies that the live in service can be doubled. However, this change is sensitive to the testing conditions, i.e., to the service conditions, and suggests that hardness is not the key factor that defines the wear rate of the different materials and conditions tested.

  1. Inclusion of Wear Debris Outcome in Conclusion:
    • The conclusion does not mention the outcomes from the wear debris (presented in Fig 13). Can this be addressed?

Following the reviewer suggestion, a new conclusion has been added:

  • The wear debris formed during the tests consisted of material from the ZX11 alloy and the AZ31 counterbody. The first one shows an irregular morphology and it has a higher O content, being related to the oxide layer formed during the wear test. The second one is more elongated and exhibits higher Al content, which corresponds to that coming from the AZ31 alloy counterbody. Material transfer from the counterbody was particularly relevant in the heat-treated conditions.
  1. Enhancement of Conclusion:
    • The conclusion could be improved by specifying the key outputs or findings of the manuscript.

An additional conclusion has been added.

  1. Inclusion of Additional Relevant Papers:
    • It would be beneficial if the authors could include references to papers related to the processing and wear mechanisms of magnesium alloys.

The references suggested by the reviewer have been added to the text.

We acknowledge the reviewer for the detailed analysis made to improve our work.

Round 2

Reviewer 2 Report

Comments and Suggestions for Authors

All comments and issues were addressed adequately.

Author Response

We thank the reviewer for the quick answer.

Reviewer 4 Report

Comments and Suggestions for Authors

The authors have responded to most of the comments.

Some comments need to be revisited for further improvement such as: 

Table 1 mentioned in the response is not available in the manuscript. 

 Authors have uploaded the manuscript at place of authors response. 

Comments on the Quality of English Language

The authors have responded to most of the comments.

Some comments need to be revisited for further improvement, such as:

Is EDS analysis alone sufficient to accurately predict elemental composition?

Which wear mechanisms are prevalent under 2N and 20N loads?

etc.

Table 1, mentioned in the response, is unavailable in the manuscript.

The authors have uploaded the manuscript in place of the authors response.

Author Response

Table 1, mentioned in the response, is unavailable in the manuscript.

Table 1 has not been included in the manuscript because only some of the heat treatments included in it have been used in the present study. Moreover, permission is not available to reproduce this table in this manuscript. However, the basic temperature designations and the reference have been included in the manuscript.

Is EDS analysis alone sufficient to accurately predict elemental composition?

In this study, the EDS analysis is sufficient to compare the material from the pin and the counterbody due to the presence of Al in the latter. It has result to be a useful tool to compare different samples at a compositional level in this study. To accurately predict elemental composition other analytical techniques could be included, but we consider that it would not be necessary to meet the objectives of this study.

Which wear mechanisms are prevalent under 2N and 20N loads?

As mentioned before, no exclusive wear mechanisms have been observed for each of the loads used. However, the presence of oxidation was more pronounced in the samples tested at 20N, as shown in the manuscript. Additional information about wear mechanisms can be found throughout the “3. Results and discussion - 3.5. Wear Mechanisms” section of the manuscript.

We thank the reviewer again for the analysis carried out to improve our work.